# Earth's magnetosphere and outer radiation belt under sub-Alfvénic solar wind

Noé Lugaz[1,2], Charles J. Farrugia[1,2], Chia-Lin Huang[1,2], Reka M. Winslow[1], Harlan E. Spence[1,2] & Nathan A. Schwadron[1,2]

The interaction between Earth's magnetic field and the solar wind results in the formation of a collisionless bow shock 60,000–100,000 km upstream of our planet, as long as the solar wind fast magnetosonic Mach (hereafter Mach) number exceeds unity. Here, we present one of those extremely rare instances, when the solar wind Mach number reached steady values <1 for several hours on 17 January 2013. Simultaneous measurements by more than ten spacecraft in the near-Earth environment reveal the evanescence of the bow shock, the sunward motion of the magnetopause and the extremely rapid and intense loss of electrons in the outer radiation belt. This study allows us to directly observe the state of the inner magnetosphere, including the radiation belts during a type of solar wind-magnetosphere coupling which is unusual for planets in our solar system but may be common for close-in extrasolar planets.

[1] Space Science Center, Institute for the Study of Earth, Oceans, and Space, University of New Hampshire, 8 College Road, Durham, New Hampshire 03824, USA. [2] Department of Physics, University of New Hampshire, Durham, New Hampshire 03824, USA. Correspondence and requests for materials should be addressed to N.L. (email: noe.lugaz@unh.edu).

Under normal solar wind conditions, a bow shock forms sunward of Earth with a subsolar distance of 11–14 Earth radii ($R_E = 6,371$ km). Behind it, in the magnetosheath, the flow becomes slower than the fast magnetosonic speed. The magnetosheath is separated from the magnetosphere by the magnetopause, with a typical subsolar distance of 9–11 $R_E$. Inside the magnetosphere, the radiation belts extend from $\sim 2$–7 $R_E$ and contain trapped electrons with energies between few hundreds of keV and several MeV. The shape and location of these boundaries are strongly determined by the solar wind Mach number and dynamic pressure as well as the interplanetary magnetic field[1,2]. In the solar wind upstream of Earth, the Mach number is typically in the (2, 10) range, with the lowest values reached during coronal mass ejections (CMEs). Major efforts have been undertaken to characterize the coupling between Earth's magnetosphere and the solar wind under different Mach regimes[3,4]. Sub-fast solar wind conditions (when the solar wind speed is lower than the fast magnetosonic speed) represent the most extreme and unusual regime which occurs at Earth (on average, only 2–3 times per decade, see ref. 5). Numerical studies have shown that, when the Mach number drops below 1, Earth's magnetosphere dramatically changes and forms so-called Alfvén wings[6,7]. This has been confirmed by a few reported detections at Earth[5,8]. However, none of these detections included information about Earth's radiation belts or the dayside magnetosphere. Although extremely rare at Earth, sub-fast stellar winds may be common for extrasolar planets close to their host stars[9,10]. Understanding the response of the Earth's magnetosphere and radiation belts to sub-fast solar wind can therefore bring new insight into the type of radiation environment around extrasolar planets with intrinsic magnetic fields.

On 13 January 2013, a slow CME erupted from the Sun with a speed of about 400 km s$^{-1}$. Its appearance as a faint partial halo in SOHO LASCO/C2 instrument[11] and as a limb event for the two STEREO/SECCHI/COR2 instruments[12] confirm that it was Earth-directed (See Supplementary Fig. 1 and Supplementary Note 1). The Wind spacecraft was far upstream from Earth (195 $R_E$) and provided interplanetary monitoring of this event, as shown in Fig. 1. A perturbation characteristic of the *in situ* counterpart of a CME can be identified around 00:00 universal time (UT) on 17 January 2013, and is characterized, first, by elevated and highly variable pressures for 16 h, with rapid changes occurring at field and flow discontinuities. The interplanetary magnetic field, initially northward in the GSM coordinate system, turns southward at 13:00 UT, which results in the main phase of a moderate geomagnetic storm. The CME magnetic ejecta starts at 16:00 UT, as evidenced by a smooth and elevated magnetic field and large rotations of the field. Simultaneously, the proton number density drops precipitously, by two orders of magnitude, reaching values well below 1 cm$^{-3}$ by 19:00 UT (see Supplementary Note 2). The proton density remains extremely low until 02:00 UT on 18 January, whereas the magnetic ejecta ends at 12:00 UT on the same day. From 18:15 UT to 23:50 UT on 17 January, the solar wind Mach number remains below 1, except for a short ($\sim 30$ min) interruption of Mach $\sim 2$–3 around 20:30 UT.

During this period of sub-Alfvénic and sub-fast solar wind, more than 10 spacecraft made measurements in the dayside magnetosphere and in Earth's radiation belt. Even during periods of southward interplanetary magnetic fields, the geo-effects induced by this CME were only moderate. However, this period was also characterized by a long-lasting electron dropout in the Earth's radiation belts and large oscillations in the magnetospheric magnetic field.

## Results

**Bow shock and magnetopause locations.** To determine the location of the bow shock and magnetopause, we used a combination of satellite crossings and typical models for their location[1,2] as shown in Fig. 2. With the arrival of the two solar wind dynamic pressure increases at 00:00 UT and 12:00 UT on 17 January, the magnetopause and bow shock moved Earthward to a minimum distance of 6.28 $R_E$ and 8.15 $R_E$, respectively, around 13:40 UT. Consequently, from 13:15 UT to 15:20 UT on 17 January, ten magnetospheric spacecraft (Cluster, THEMIS, ARTEMIS-2 and Geotail) were in the solar wind, measuring the same plasma as Wind (see Supplementary Fig. 3 for the orbit of some of the spacecraft in this study). This rare occurrence of so many near-Earth spacecraft being simultaneously in the solar wind is due to the strong dynamic pressure and southward interplanetary magnetic field at the time. The field in the dense region before the magnetic ejecta was planar with significant non-radial velocity components (see Supplementary Note 3). This type of planar structure is often seen in the sheath region ahead of a magnetic ejecta[13,14], suggesting that, even in the absence of a

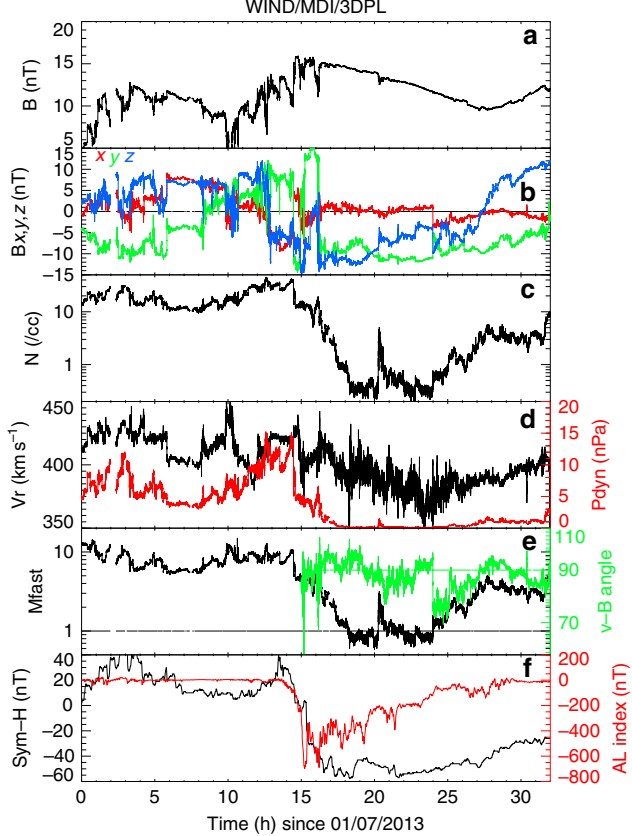

**Figure 1 | Solar wind measurements and geomagnetic indices on 17–18 January 2013.** The solar wind measurements are made by the Wind spacecraft upstream of Earth's magnetosphere. The panels show the total magnetic field (**a**), the magnetic field components in GSM coordinates (**b**; x component in red, y component in green and z component in blue), the number density (**c**), the radial velocity and dynamic pressure (in red; **d**) (**d**), the fast magnetosonic Mach number and angle between the magnetic field and velocity vectors (in green; **e**), the Sym-H index and the AL index (in red; **f**). The Sym-H is a 1-minute index which characterizes the disturbance of mid-latitude geomagnetic field, whereas the AL is a 1-minute index which characterizes the auroral activity in the northern hemisphere. The dense sheath preceding the magnetic ejecta starts around 00:00 UT on 17 January, the ejecta at 16:00 UT and the sub-fast period from 18:15 UT to 23:50 UT, except for a 30-minute period of Mach $\sim 2$ centred around 20:30 UT.

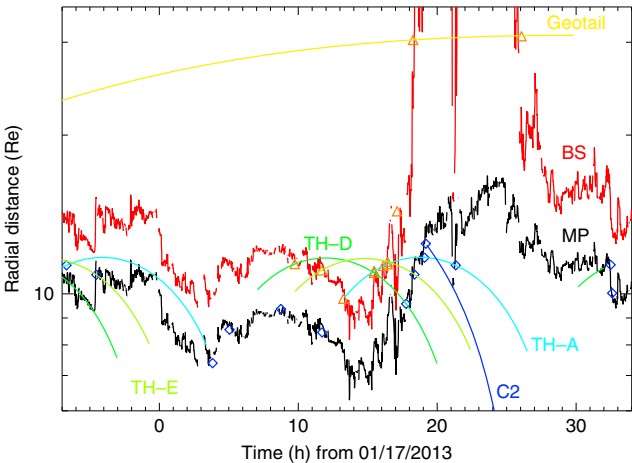

**Figure 2 | Measured and modeled magnetopause (black) and bow shock (red) nose locations.** The magnetopause model is that of Shue et al.[1], the bow shock that of Farris et al.[2]. Blue diamonds and orange triangles mark spacecraft crossings of these boundaries (TH: Themis, C2: Cluster-2). The radial distance of the spacecraft when they are within ±25° of the Sun-Earth line are drawn, showing the locations where they give accurate estimate of the location of the boundaries. From hours 13.25 to 15.30, all spacecraft used in the study were in the solar wind. The large sunward motion of the magnetopause and bow shock started at 16 h and the sub-fast period at 18.25 h. During the short period of superfast solar wind at 20.5, the magnetopause crossed Themis-A's location.

CME-driven shock, the dense region starting at 00:00 UT is being formed by the accumulation of material due to the propagation of the magnetic ejecta through the solar wind. The pressure tensor (see Supplementary Note 3) shows that there were significant transverse stresses being applied to the magnetosphere. From 08:00 UT to 12:30 UT, the dynamic pressure steadily increased to reach values around 15 nPa. Following a drop to 9 nPa, the normal and tangential pressures increased gradually starting at 13:30 UT and abruptly dropped at 14:28 UT.

As the dynamic pressure and the solar wind Mach number suddenly decreased at the arrival of the magnetic ejecta, the bow shock location can be tracked outward through its crossings of the Cluster and THEMIS spacecraft, first, and then of ARTEMIS-2 and Geotail (see Fig. 2). The nose of the bow shock went from $\sim 11\ R_E$ (last Cluster crossing) at 17:10 UT to $\sim 30\ R_E$ (Geotail crossing) at 18:20 UT, moving with an average speed of $30\ \mathrm{km\ s^{-1}}$. Eventually, the bow shock crossed ARTEMIS-2 at a location of $58.8\ R_E$ in the dayside flank, a distance similar to that of the farthest Earth's bow shock dayside crossing ever reported, on 11 May 1999 (aka the 'day the solar wind almost disappeared')[15].

The initial drop in the pressures exerted on the magnetopause at the start of the magnetic ejecta at 14:28 UT was extremely abrupt with a decrease from $\sim 15$ nPa to 5 nPa in the normal pressure in $<30$ s. This sharp discontinuity should result in a rapid sunward acceleration of the magnetopause[16]. Shortly after crossing the bow shock into the magnetosheath, the Cluster and THEMIS spacecraft crossed the magnetopause into the magnetosphere (see Supplementary Note 4). The crossings imply that the magnetopause moved outward from 9.5 to $12.5\ R_E$, from 17:45 to 19:10 UT (see Fig. 2). The nose of the magnetopause may have reached distances of $16\ R_E$ during this period according to the model of Shue et al.[1], although the model may not be applicable to sub-fast solar wind conditions (see Supplementary Note 5). Overall, the magnetosphere was found to 'bulge out' similar to previous reports[8,17].

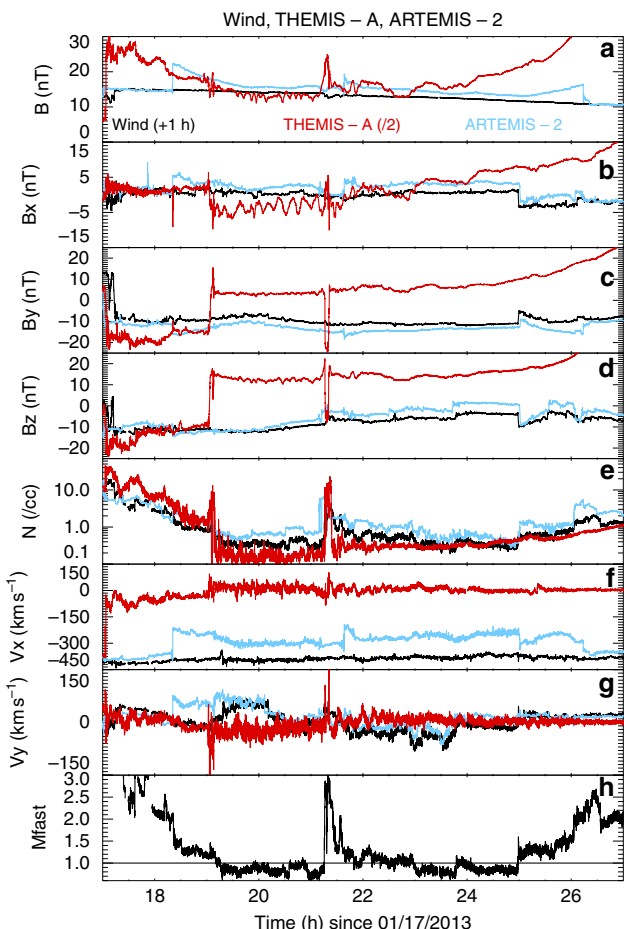

**Figure 3 | Dayside magnetospheric and solar wind measurements.** The magnetospheric measurements are taken by ARTEMIS-2 (blue) and THEMIS-A (red, magnetic field divided by 2) and the solar measurements by Wind (black, shifted by one hour). The panels show the magnetic field strength (**a**), the x, y and z component of the magnetic field (**b-d**), the proton number density (**e**), the x and y components of the velocity (**f** and **g**) and the fast magnetosonic Mach number (**h**). ARTEMIS-2 data shows an inbound bow shock crossing at 18.25 h, followed by small increases in magnetic field and small decreases in velocity. (ii) THEMIS-A data shows the magnetopause crossing at around 19:00 UT, followed by a boundary layer with large increases in the y components of the magnetic field and the velocity, and further followed by large oscillations in the magnetic field for several hours. At around 21:20 UT, ARTEMIS measured the reformation of the bow shock, while THEMIS-A crossed the magnetopause during the short period of superfast flows.

During the short period of 30 min starting at 21:00 UT when the solar wind Mach number became $>1$, the bow shock immediately reformed and was crossed moving outward by ARTEMIS-2 at the end of the period. Simultaneously, THEMIS-A crossed the magnetopause into the magnetosheath. This implies that the bow shock reformed earthward of Geotail, that is, between 12 and $30\ R_E$.

During the prolonged sub-fast period, there are strong indications that ARTEMIS-2 was not in the magnetosheath but rather in the solar wind disturbed by Earth, as it witnessed flows and magnetic fields slightly different from that of the far upstream solar wind (Fig. 3). We discounted the hypothesis that these slightly different magnetic field measurements are due to a lunar wake effect, since Geotail observes very similar magnetic field temporal profiles to ARTEMIS-2 and Wind (see Supplementary Fig. 6). At ARTEMIS-2, the flow speed was

as fast as 310 km s$^{-1}$, primarily in the GSM $x$ direction, and the density was comparable to that of the solar wind. The magnetic field strength was only 10% larger than that measured by Wind with an almost identical value of $B_z$ (from 21:00 UT onward) component but positive values of $B_x$. These measurements are consistent with a slowing down of the solar wind and draping of the interplanetary magnetic field on approach to Earth's magnetosphere. This is the expected behaviour for the interaction of a sub-fast solar wind with a magnetized obstacle. Interesting periods are also observed when ARTEMIS-2 was in the magnetosheath under a weakly superfast solar wind just before and after the sub-fast period (from 18:20 to 19:10 on 17 January and from 01:00 to 02:15 on 18 January). The measurements at these times were somewhat unexpected, as the magnetic field rotated without changing its magnitude, and ARTEMIS-2 measured denser and faster plasma.

During the sub-fast period, all THEMIS and Cluster spacecraft were in the magnetosphere, whereas Geotail and ARTEMIS were in a flow consistent with the solar wind modified by the presence of the Earth's magnetosphere, but not indicative of Alfvén wings as in past work[8]. The interplanetary magnetic field was primarily in the $-y$, $-z$ direction and perpendicular to the solar wind velocity (as shown in the green line in the penultimate panel of Fig. 1), resulting in wings extending anti-sunward (in the nightside). Geotail and ARTEMIS are therefore expected to be both outside of the wings.

**Geomagnetic response.** The solar wind dawn-to-dusk electric field was at ~4 mV m$^{-1}$ for >3 h around 18:00 UT, a condition which is typically associated with intense geomagnetic storms (Dst below −100 nT) as defended by Gonzalez and Tsurutani[18]. This strong driving is corroborated by THEMIS and Cluster measurements at the magnetopause (see Fig. 3 for THEMIS-A measurements). These confirm that the magnetic field clock angle through the boundary rotated by more than 135°, conditions typically associated with strong magnetic reconnection (for example, see ref. 19). In fact, as the Mach number is low, the reconnection rate at the magnetopause is expected to be high. However, during this period, there was only a moderate storm (minimum Dst at −53 nT from 23:00 UT on 17 January to 01:00 UT on 18 January) with relatively weak substorms, as indicated by the AL index reaching values in the 500–700 nT range. This appears to confirm previous findings that sub-fast solar wind conditions result in a quietening of global magnetospheric disruption[8,20].

The situation here is, however, significantly more complex when we consider measurements made by ground magnetometers such as those from the IMAGE magnetometer chain, and in the dayside magnetosphere. In fact, large-amplitude (few hundred nT) oscillations of the magnetospheric magnetic field as measured from the ground started around 15:00 UT, that is, around the time that the first drop in pressures reached the magnetopause (see Supplementary Fig. 9). Its start is well synchronized with an abrupt drop in the normal and tangential stresses at 14:28 UT. These oscillations continued until minimum densities were reached, to be replaced by other strong oscillations lasting this time until 05:00 UT on 18 January. As the THEMIS and Cluster spacecraft crossed into the magnetosphere, they measured similar large-amplitude oscillations in the magnetic field, primarily in the $B_x$ and $B_z$ components, reaching amplitudes of 10 nT ($\Delta B/B \sim 10$–15%) and having varying periods between 5 and 15 min (see Supplementary Figs 6 and 7). We hypothesize that these oscillations are due to ultra-low frequency waves, related to the extreme decrease in dynamic pressure at the nose of the magnetopause, similar to what has been described in recent studies[21,22].

**Radiation belt measurements.** Measurements in the radiation belts further confirm that the state of Earth's inner magnetosphere was far from typical. The Van Allen Probes (RBSP, see ref. 23) were in the dusk and night sides of the radiation belt, and the GOES satellite in the dawn and day sides, providing us with the first detailed measurements of Earth's radiation belts during a prolonged interval of sub-fast solar wind. Previous work by Goldstein et al.[24] focused on the formation of plasmapause plumes during this time period, and indicated that the Van Allen Probes were not in a plume in the period of interest.

A significant dropout of the flux of energetic electrons above 200 keV was recorded by the Van Allen Probes as shown in Fig. 4 (GOES also recorded a dropout of electrons above 800 keV, as shown in the last panel of this Figure). It starts when the Van Allen Probes entered the outer radiation belt around 18:00 UT on 17 January. All energies (above 200 keV) and all pitch angles were affected, indicating that these dropouts corresponded to a general loss of energetic electrons, presumably into the solar wind, consistent with one of the major causes of electron dropouts[25]. The dropout of energetic electrons of energies above 1 MeV reached L-shells of 4 $R_E$, almost fully emptying the outer radiation belt. The effects only lasted for one RBSP orbit for low energies, corresponding to the duration of the low-Mach number solar wind, but the decreased fluxes of high energy electrons (MeV and higher) persisted for several days at high L-shells (L>5), as shown in Fig. 5. To fully quantify the loss of energetic electrons, it would be necessary to derive the particle phase space density, which would require calculating the L* shell parameter[26]. During this period of very low dynamic pressure and sub-fast flows, such a calculation would have severe limitations and unknown uncertainties. However, Figure 5 illustrates how the flux of energetic electrons at high L-shells has been lowered by an order of magnitude from early 17 January to 18 January and low fluxes persisted for several days afterwards.

We now discuss the causes of this electron dropout. As is clear from Fig. 4, there are no energetic electron losses at geosynchronous orbit from 08:00 to 15:00 UT on 17 January, a time period when the solar wind dynamic pressure increased from 4 to 15 nPa. In fact, there is a small intensification of highest energy electrons as measured by RBSP/REPT and GOES-13 at energies above 2 MeV (see Supplementary Note 6 and Supplementary Fig. 10). We hypothesize that the following scenario took place: (1) the large dynamic pressure with large tangential stresses from 00:00 UT to 14:00 UT pushed earthwards and distorted the magnetosphere; although the pressure reached its maximum from 12:00 to 14:00 UT, there were no significant losses of energetic electrons at these times. (2) At 14:28 UT, there was an abrupt drop in dynamic pressure. This relaxed the stresses (both normal and tangential) and resulted in a rapid sunward motion of the magnetopause[27], a global response was seen in ground magnetometers and a large deformation of the magnetopause was observed. In the following hours, the dynamic pressure continued to decrease to reach extremely low values and the solar wind flow became sub-fast. This resulted in a continued inflation of the magnetosphere, a disappearance of the bow shock and the continuation of waves being transferred into the magnetosphere as measured by THEMIS and Cluster. These ultra-low frequency waves with large amplitude may have strongly enhanced the outward radial diffusion of energetic particles following the mechanism proposed by Ukhorskiy et al.[28,29]. The energetic particle losses started with the pressure drop and continued during this time. (3) As the solar wind flow became superfast starting around 00:00 UT on 18 January, there was a recovery of energetic electrons as measured by GOES and RBSP, although the highest energy electrons (with energies above 4 MeV) did not recover for several days at L-shells >4.5.

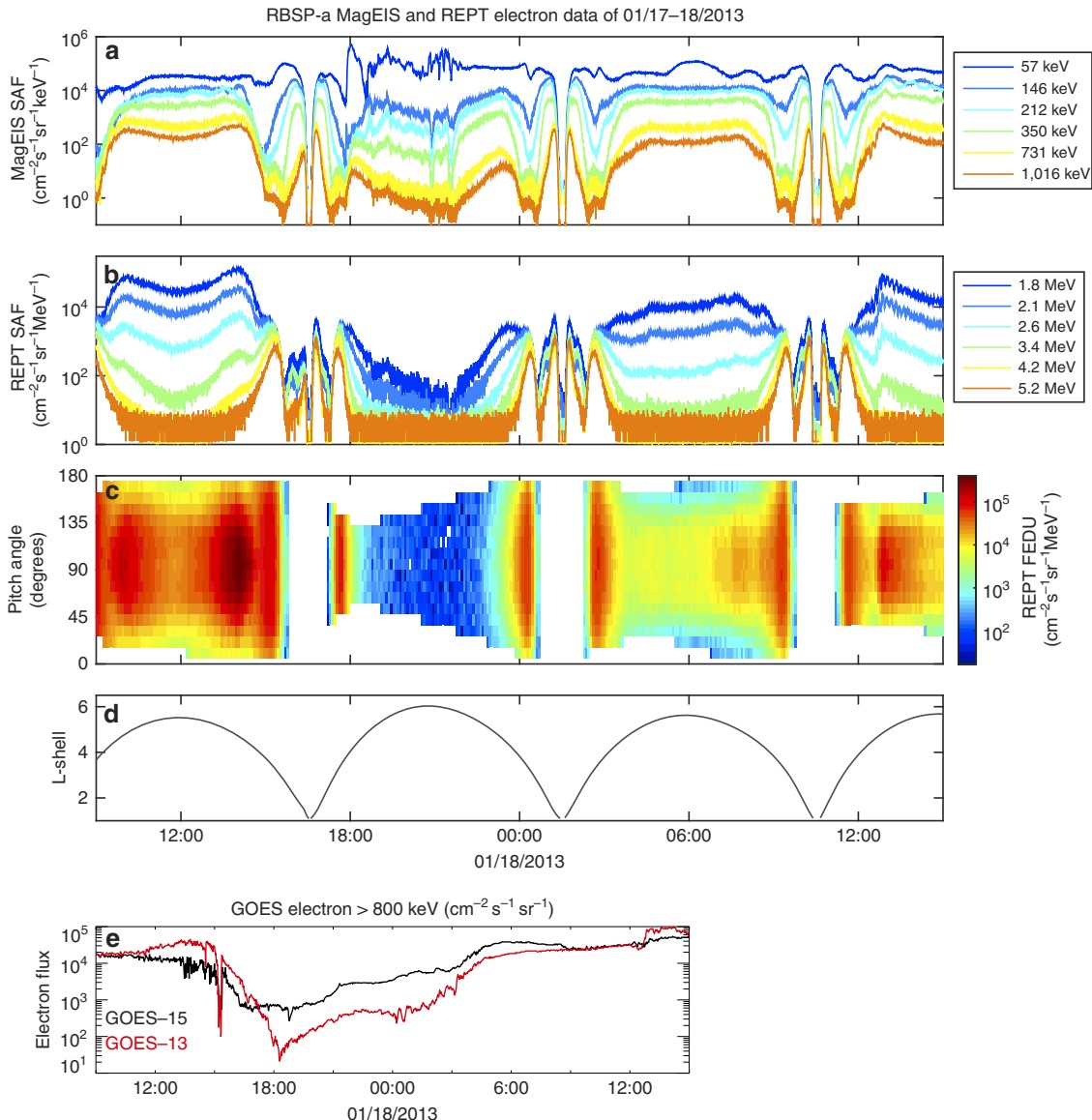

**Figure 4 | Measurements of energetic electrons in the radiation belts.** (**a**) shows RBSP-A/ECT/MagEIS electron fluxes (between 50 keV and 1 MeV), (**b**) RBSP-A/REPT electron fluxes (between 1.8 MeV and 5.2 MeV), (**c**) electron fluxes at 2.6 MeV sorted by pitch angles and (**d**) L-shell associated with RBSP-A orbit. (**e**) shows electron flux measurements (energies above 800 keV) by GOES-13 and 15 at geosynchronous orbit. The dropout of energetic electrons above 200 keV in the outer radiation belt is clearly seen in RBSP-A measurements starting around 18:00 UT, it occurs at all pitch angles, indicating this is not related to a loss to the atmosphere. GOES measures a loss staring around 15:00 UT and lasting throughout the sub-fast period.

Because at the time of maximum dynamic pressure, GOES was in the dayside and the Van Allen Probes did not enter the outer radiation belt until 18:00 UT, we cannot rule out that earlier losses may have occurred without being measured. The scenario we proposed here differs in some respects from the 'traditional' magnetopause shadowing[25,30,31], when the earthward motion of the magnetopause during periods of high dynamic pressure results in the opening of previously closed drift shells and the subsequent loss of energetic electrons, previously trapped on these orbits. One likely explanation for the difference is that, on 17 January 2013, the dynamic pressure increase was more gradual than during the passage of most CME-driven sheath regions, and the outer belt electrons may have been able to respond adiabatically.

## Discussion

We have identified a period of sub-Alfvénic and sub-fast solar wind during which more than 10 spacecraft made measurements in the dayside magnetosphere and in Earth's radiation belt. Although the solar wind forcing would have predicted an intense geomagnetic storm, the geo-effects as monitored by global indices were only moderate. Two other unusual phenomena occurred: a long-lasting electron dropout in the Earth's radiation belts and large oscillations in the magnetospheric magnetic field. These show that this period is far from being quiet in terms of geo-effects in the inner magnetosphere, contrary to the 'day the solar wind almost disappeared'[20] and to most previous instances of sub-fast solar wind. One reason is certainly that the interplanetary magnetic field had a strong GSM southward component, the opposite of both cases recently simulated by Chané *et al.*[7].

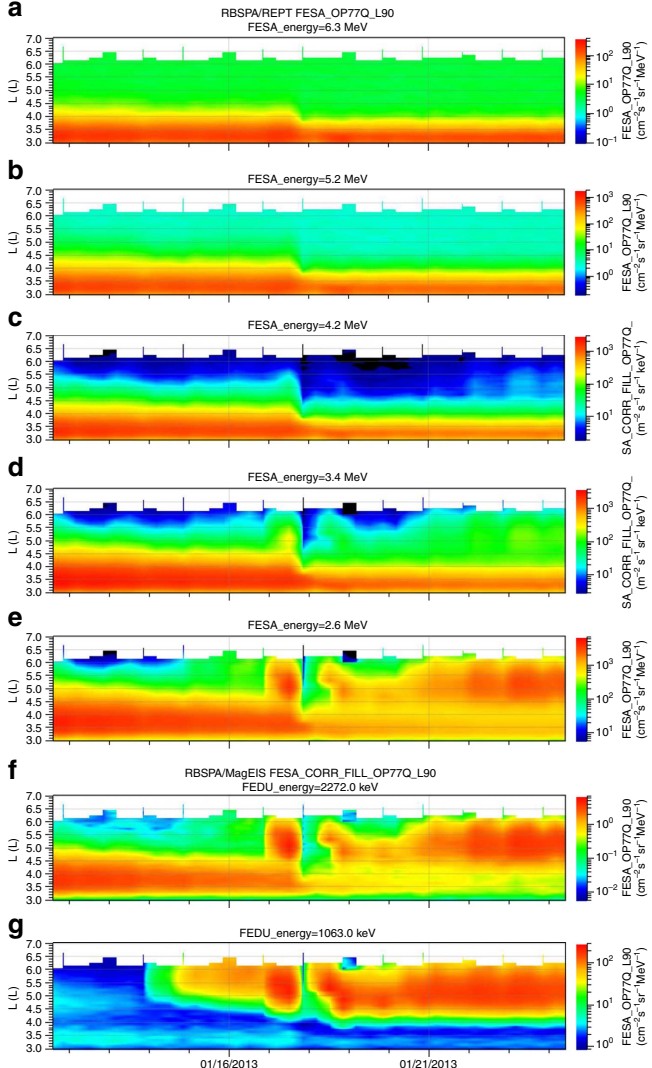

**Figure 5 | Long-term measurements of relativistic electrons in the radiation belts.** The measurements are made by RBSP-A/ECT showing the REPT (**a**–**e**) and MagEIS (**f** and **g**) electron fluxes plotted against the L-shell for a 13-day period centred on 18 January. The fluxes are in unit of particles $s^{-1}\,sr^{-1}\,cm^{-2}\,MeV^{-1}$ for REPT and $s^{-1}\,sr^{-1}\,cm^{-2}\,keV^{-1}$ for MagEIS. The median energy of the electrons for each plot is shown on the subtitle of each plot and vary from 6.3 MeV at the top to 1.06 MeV at the bottom. The loss of energetic electrons late on 17 January can be clearly identified and its effects (lower fluxes at 2 MeV and above) lasted more than a week.

Cohen et al.[10] studied the effects on a planetary atmosphere of stellar winds transitioning from superfast to sub-fast. Other researchers have also discussed the ubiquity of radiation belts in our solar system and their likely existence in other solar systems[32]. To study how a planet's heliocentric distance influences the frequency of sub-fast events, we have analysed all magnetometer data from the MESSENGER orbital mission at Mercury, which lasted from March 2011 to April 2015. We have identified nine candidate sub-fast events, that is, events for which there was no bow shock crossing over at least one orbit around Mercury. We followed the procedure of Winslow et al.[33] to identify bow shock crossings. At Mercury's mean orbital distance of 0.39 AU, we find a rate of sub-fast events of about 2.2 events/year, about ten times higher than the rate at 1 AU. We can therefore safely conclude that many close-in planets orbiting

solar-type stars would encounter sub-fast conditions several times per year, strongly affecting any potential outer radiation belts (in the present study, the inner radiation belt remains overall unaffected). This would occur even when the planetary orbit is well outside the Alfvénic surface. In many cases, these sub-Alfvénic periods will only be temporary as shown in Cohen et al.[10]. It should also be noted that the conditions found in the present study are very different from those expected for many extrasolar systems (for example, see ref. 34) and there are certainly a wide variety of star-planet interactions, depending on the strength of the stellar wind, the magnetic field of the star and that of the planet, and the planet's orbit. In some cases, a planet may be under steady sub-fast stellar winds, a situation on which this study cannot shed light. However, this event provides the first *in situ* measurements of the radiation belt of any planet under sub-fast conditions.

**Data availability**. The THEMIS, ARTEMIS, Wind, ACE, Cluster, Geotail and GOES data that support the findings of this study are available from CDAWeb from NASA/GSFC Space Physics Data Facility (http://cdaweb.gsfc.nasa.gov/istp_public/). The Van Allen Probes data that support the findings of this study are available from the RBSP/ECT Science Operations and Data Center maintained by the Los Alamos National Laboratory (http://www.rbsp-ect.lanl.gov). The STEREO and SOHO data that support the findings of this study are available from the SECCHI Naval Research Laboratory database (http://secchi.nrl.navy.mil) and LASCO Naval Research Laboratory database (http://lasco-www.nrl.navy.mil/). The IMAGE ground magnetometer network data that support the findings of this study are available from the IMAGE database maintained by the Finnish Meteorological Institute (http://space.fmi.fi/image/beta/).

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

## Acknowledgements

We acknowledge the use of data from THEMIS, ARTEMIS, the Van Allen Probes, Wind, ACE, Cluster, Geotail, STEREO/SECCHI, SOHO/LASCO, the IMAGE ground magnetometer network and GOES. We thank the institutes who maintain the IMAGE Magnetometer Array. We thank the reviewers for their useful comments which improved both the presentation and scientific content of the article. This work is supported by NASA grants NNX15AB87G, NNX13AI75G and NNX13AH94G and NSF grants AGS1433213 and AGS1435785, as well as RBSP-ECT funding provided by JHU/APL contract 967399 under NASA's Prime contract NAS5-01072.

## Author contributions

N.L. identified the event, performed the initial analyses of spacecraft data, coordinated the study and wrote the initial draft of the manuscript. C.J.F. gave input on the analysis of measurements by Wind, Geotail, ARTEMIS, Cluster and THEMIS. C.-L.H. gave input on radiation belt measurements by RBSP and measurements by THEMIS and prepared (Fig. 4). R.M.W. determined the sub-Alfvénic rate at Mercury and gave input about the relevance to exoplanets. N.A.S. and H.E.S. participated in the discussion and gave many suggestive comments.

## Additional information

**Competing financial interests:** The authors declare no competing financial interests.

