## [Peer Review File · Nature Communications]

Editorial Note: This manuscript has been previously reviewed at another journal that is not operating a transparent peer review scheme, whose name has been redacted. This document only contains reviewer comments and rebuttal letters for versions considered at Nature Communications.

PEER REVIEW FILE

Reviewers' comments:

Reviewer #2 (Remarks to the Author):

Lugaz et al. present an interesting case study of an unusual period of activity in Earth's magnetosphere in which the solar wind was sub-Alfvénic. They focus on the unprecedented multipoint satellite coverage during this unusual event, the effect on Earth's outer radiation belt electrons, and present implications for exoplanetary systems.

I served as one of the reviewers for the previous manuscript by these authors that was submitted to [redacted]. I find that the authors have done a nice job addressing the reviewers' comments from that version, and the current version of the manuscript has improved nicely. I thank the authors for their work on this revision and can recommend this version for publication in Nature Communications after the authors address a few minor comments, listed here.

Minor Comments:

1. Concerning the nature of the radiation belt dropout: The authors point to the GOES observations as evidence that the dropout did not start until after the pressure suddenly relaxed in the system around 15 UT. There was no clear, sharp pressure pulse in the solar wind based on Figure 1. Instead, the dynamic pressure increased in a series of smaller steps over a ~7 hour period. The effects of magnetopause shadowing as described by Turner et al. [2014] will only be "instantaneous" 1) at those L-shells immediately affected by the inward motion of the

magnetopause, and 2) only if the pressure increase is gradual enough that the radiation belt electrons do not respond adiabatically, that is while still conserving all three adiabatic invariants. So, the dropout may not have been observed at GOES during the pressure increase period for two possible reasons: 1) with GOES being on the dayside of the system, they actually map to L^* significantly lower than that when they are on the nightside. Those L^* s might not have been directly affected or even near those affected by the magnetopause shadowing. 2) The pressure increase in the solar wind might have been slow enough that the outer belt electrons were (mostly) able to respond adiabatically, which is quite consistent with the increase in fluxes shown in Figure 5 observed during the compression period on the 17th immediately before the sudden dropout. I hope the authors can revise the discussion on pages 9 and 10 to reflect these points concerning the nature of the dropout.

2. On page 11 of the revised manuscript, the authors state: "These show that this period is far from being quiet in terms of geo-effects in the magnetosphere." Just because the solar wind becomes exceptionally slow and tenuous does not mean we should expect the conditions in the magnetosphere to become "quiet". This slow and tenuous solar wind marked a very sudden, abrupt, and extreme change in conditions, so some activity in the system responding to those sudden changes should indeed be expected. That being said, I hope the authors can be careful with their implications for exoplanets. Earth's magnetosphere was not able to reach a "steady-state" under sub-Alfvénic conditions... it didn't spend days bathed in sub-Alfvénic solar wind that would allow the system to normalize to those driving conditions. Instead, it only responded drastically to a very strong change from its normal driving conditions. So, we can't really conclude anything significant about exoplanetary systems that would be under sub-Alfvénic driving conditions typically. This is particularly true about the radiation belts, since it is well known that Earth's radiation belts are the result of a delicate balance between sources and losses and vary wildly for often minor changes in the solar wind driving conditions. So we can't say anything at all about whether an outer electron radiation belt would form given quasi-steady longer term (multiple days) driving under sub-Alfvénic solar wind. The authors also fail to point out that the inner radiation belt was most likely completely unaffected by this event, meaning that such a belt could indeed exist at some exoplanetary systems regardless of whether they are super- or sub-Alfvénically driven. Both of those points should also be mentioned if the authors

insist on discussing the extrapolation of their results to exoplanetary systems.

3. In the response to reviewers, the authors stated that: "when the fast magnetosonic Mach number is below 1, the solar wind is always sub-Alfvenic..." that is not an accurate statement. The fast magnetosonic speed is the fastest of the three MHD fluid speeds in plasma physics, as acknowledged by the authors. That being true, it means that the flow of a plasma can be sub-fast while still being super-Alfvenic and super-slow (i.e., moving faster than the slow magnetosonic speed). That is, since V_{fm} (fast magnetosonic) $>$ V_A (Alfven speed) and $V_{fm} > V_{sm}$ (slow magnetosonic), there is some range of speeds, u , in which $V_{sm} < V_A < u < V_{fm}$. However, I appreciate that the authors pointed out how close the Alfvenic and fast magnetic speeds are in this event to justify their use of the term sub-Alfvenic throughout.

4. The lines in Figure 2 and Supplementary Figure 6 should be made thicker. Several are difficult to see when they are printed out, as can be a common issue with IDL plot's default line thickness.

We thank the reviewer for his very helpful comments. We tried to address them.

Minor Comments:

1. Concerning the nature of the radiation belt dropout: The authors point to the GOES observations as evidence that the dropout did not start until after the pressure suddenly relaxed in the system around 15 UT. There was no clear, sharp pressure pulse in the solar wind based on Figure 1. Instead, the dynamic pressure increased in a series of smaller steps over a ~7 hour period. The effects of magnetopause shadowing as described by Turner et al. [2014] will only be "instantaneous" 1) at those L-shells immediately affected by the inward motion of the magnetopause, and 2) only if the pressure increase is gradual enough that the radiation belt electrons do not respond adiabatically, that is while still conserving all three adiabatic invariants. So, the dropout may not have been observed at GOES during the pressure increase period for two possible reasons: 1) with GOES being on the dayside of the system, they actually map to L* significantly lower than that when they are on the nightside. Those L*s might not have been directly affected or even near those affected by the magnetopause shadowing. 2) The pressure increase in the solar wind might have been slow enough that the outer belt electrons were (mostly) able to respond adiabatically, which is quite consistent with the increase in fluxes shown in Figure 5 observed during the compression period on the 17th immediately before the sudden dropout. I hope the authors can revise the discussion on pages 9 and 10 to reflect these points concerning the nature of the dropout.

We changed the discussion of these points to reflect the reviewer's comment. We also removed two sentences discussing the lack of electron dropout during the sheath passage. This section now reads:

"Because, at the time of maximum dynamic pressure, GOES was in the dayside and the Van Allen Probes did not enter the outer radiation belt until 18UT, we cannot rule out that earlier losses may have occurred without being measured. The scenario we proposed here differs in some respects from the "traditional" magnetopause shadowing [Turner:2012, Turner:2014, Lugaz:2015b], when the earthward motion of the magnetopause during periods of high dynamic pressure results in the opening of previously closed drift shells and the subsequent loss of energetic electrons, previously trapped on these orbits. One likely explanation for the difference is that the dynamic pressure increase was more gradual than during many CME-driven sheath regions, and the outer belt electrons may have been able to respond adiabatically."

2. On page 11 of the revised manuscript, the authors state: "These show that this period is far from being quiet in terms of geo-effects in the magnetosphere." Just because the solar wind becomes exceptionally slow and tenuous does not mean we should expect the conditions in the magnetosphere to become "quiet". This slow and tenuous solar wind marked a very sudden, abrupt, and extreme change in conditions, so some activity in the system responding to those sudden changes should indeed be expected.

We added a statement about some expectations of quieting as discussed in previous

studies of sub-fast solar wind at Earth. Discussing these events with colleagues, it appears to us that most people expect that the fact that the dynamic pressure goes to ~ 0 nPa implies that the magnetosphere becomes dipole-like and quiet. Here, because B_z was south, this was not the case.

That being said, I hope the authors can be careful with their implications for exoplanets. Earth's magnetosphere was not able to reach a "steady-state" under sub-Alfvénic conditions... it didn't spend days bathed in sub-Alfvénic solar wind that would allow the system to normalize to those driving conditions. Instead, it only responded drastically to a very strong change from its normal driving conditions. So, we can't really conclude anything significant about exoplanetary systems that would be under sub-Alfvénic driving conditions typically. This is particularly true about the radiation belts, since it is well known that Earth's radiation belts are the result of a delicate balance between sources and losses and vary wildly for often minor changes in the solar wind driving conditions. So we can't say anything at all about whether an outer electron radiation belt would form given quasi-steady longer term (multiple days) driving under sub-Alfvénic solar wind. The authors also fail to point out that the inner radiation belt was most likely completely unaffected by this event, meaning that such a belt could indeed exist at some exoplanetary systems regardless of whether they are super- or sub-Alfvénically driven. Both of those points should also be mentioned if the authors insist on discussing the extrapolation of their results to exoplanetary systems.

In the previously revised version of the article, we removed most of the discussion about extrasolar planets. We feel that adding extra discussion will put, again, more emphasis on it, while it is not what is desired. We added two sentences in accordance with the reviewer's suggestions, in addition to the previous statement limiting the generality of the present study.

The end of the article now reads:

"We can therefore safely conclude that many close-in planets orbiting solar-type stars would encounter sub-fast conditions several times per year, strongly affecting any potential outer radiation belts (in the present study, the inner radiation belt remains overall unaffected). This would occur even when the planetary orbit is well outside the Alfvénic surface. In many cases, these sub-Alfvénic periods will only be temporary as shown in Cohen *et al.* (2014) \citep[Cohen:2014].

It should also be noted that the conditions found in the present study are very different from those expected for many extrasolar systems \citep[e.g., see][Vidotto:2013], and there are certainly a wide variety of star-planet interactions, depending on the strength of the stellar wind, the magnetic field of the star and that of the planet, and the planet's orbit. In some cases, a planet may be under steady sub-fast stellar winds, a situation on which this study cannot shed light. However, this event provides the first *in situ* measurements of the radiation belt of any planet under sub-fast conditions."

3. In the response to reviewers, the authors stated that: "when the fast magnetosonic Mach number is below 1, the solar wind is always sub-Alfvénic..." that is not an accurate

statement. The fast magnetosonic speed is the fastest of the three MHD fluid speeds in plasma physics, as acknowledged by the authors. That being true, it means that the flow of a plasma can be sub-fast while still being super-Alfvenic and super-slow (i.e., moving faster than the slow magnetosonic speed). That is, since V_{fm} (fast magnetosonic) $> V_A$ (Alfven speed) and $V_{fm} > V_{sm}$ (slow magnetosonic), there is some range of speeds, u , in which $V_{sm} < V_A < u < V_{fm}$. However, I appreciate that the authors pointed out how close the Alfvenic and fast magnetic speeds are in this event to justify their use of the term sub-Alfvenic throughout.

Indeed. Sorry about this confusion. The flow could in theory be sub-fast and super-slow/super-Alfvenic. In low beta plasma, as discussed in our previous answer, all three speeds are very close and the flow is typically sub-Alfvenic and sub-fast (and also sub-slow). Note that the text uses sub-fast almost throughout, except in the title and part of the conclusion.

4. The lines in Figure 2 and Supplementary Figure 6 should be made thicker. Several are difficult to see when they are printed out, as can be a common issue with IDL plot's default line thickness.

Thank you

Reviewers' Comments:

Reviewer #2 (Remarks to the Author)

I thank the authors for addressing my comments. I find the revised manuscript acceptable for publication.